# Deflection-Based Approach for Flexible Pavement Design in Thailand †

Auckpath Sawangsuriya [1,*], Tunwin Svasdisant [2] and Poranic Jitareekul [3]

1   Bureau of Road Research and Development, Department of Highways, Bangkok 10400, Thailand
2   Bureau of Highways Maintenance Management, Department of Highways, Bangkok 10400, Thailand; tunwin@doh.go.th
3   Bureau of Materials, Analysis and Inspection, Department of Highways, Bangkok 10400, Thailand; poranic@doh.go.th
*   Correspondence: auckpath@doh.go.th; Tel.: +66-91-455-6142
†   This paper was presented at the 5th International Conference on Transportation Infrastructures (V ICTI) in Lima, Peru in 10–13 August 2022. It has been selected for publication in this journal.

**Abstract:** The Department of Highways (DOH), Thailand, has adopted both empirical and mechanistic approaches for flexible pavement analysis and design. Recently, the deflection-based design approach has been comprehensively reviewed by the DOH for the possible adoption of national design standards and practices. One of the key reasons is that Thailand's road authorities, i.e., the DOH and the Department of Rural Roads (DRR), have considered the falling weight deflectometer (FWD) for the new construction and rehabilitation of road pavements. In addition, the FWD is widely accepted as the non-destructive test for deflection measurement and structural capacity evaluation. Ultimately, the implication of FWD deflections for in-house pavement analysis and design shall be developed and proposed to Thailand's road authorities. Therefore, this study presents the deflection-based approach of flexible pavement design in Thailand. The FWD and a standard Thai truck were selected as the main loading applications in this study. A typical FWD loading stress of 700–800 kPa was practically adopted by the DOH and compared with a standard 10-wheel 25-ton truck with a tandem axle-dual wheel configuration with a tire pressure of 690 kPa. The layered elastic analysis was performed to calculate the pavement responses. The results suggest that the flexible pavement design based on a deflection-based approach is simple, practical, and conservative.

**Keywords:** pavement design; surface deflection; flexible pavement; falling weight deflectometer; multi-layered elastic analysis

## 1. Introduction

In recent years, road pavement design has evolved from a purely empirical approach to mechanistic–empirical methods, which require comprehensive knowledge of material behavior and their responses, e.g., stress, strain, and deflection, under traffic loads. The newly evolved mechanistic–empirical design method involves the physical relationship between causes (e.g., traffic loads and volumes, material properties, environmental conditions, etc.) and effects (e.g., stresses, strains, deflections, etc.) in conjunction with developed mathematical models and experimental data to relate these effects to failure or distress modes. Finite element analysis (FEA) and layered elastic analysis (LEA) are commonly accepted in pavement analysis and design. FEA has been widely adopted in the recently developed mechanistic design and performance analysis of pavement systems because of its versatile implication of mechanical characterization [1–5]. LEA is practically adopted, however, because of its simplicity and cost of analysis. Both FEA and LEA assume pavement layers to be homogenous, isotropic, and linear elastic and to have finite thicknesses with a modulus of elasticity and Poisson's ratio.

In the FWD test, the pavement surface is subjected to an impulse load by dropping a mass onto a metal plate with a rubber seal placed between the plate and pavement surface to prevent the direct impact of the load. A series of sensors are installed to measure the surface deflections directly below the plate at different radial offsets. Data from the FWD are used to perform back-calculation analysis. Back-calculation is one of several techniques that are used to obtain the elastic moduli of pavement layers from the deflection basin of the FWD test. There are two approaches to back-calculation analysis, e.g., forward and backward. In the forward analysis, deflections are determined through structural analysis using information such as pavement layer thicknesses, initial mechanical properties, loading characteristics, etc. Layered elastic analysis, the finite element method, and the finite difference method are used to calculate deflections. In backward analysis, the calculated deflections obtained from the forward analysis are compared with measured deflections from the FWD test. It is noteworthy that the elastic theory exhibits some limitations especially for asphalt materials because of their viscoelastic nature. There have been attempts to consider this aspect in pavement design and maintenance. The FWD is indeed a tool that is used for many pavement applications, especially in terms of monitoring aspects. Although there are also approaches to combine FWD testing with viscoelastic analysis [6–8], such aspects were not the main aim of the study.

The Department of Highways (DOH), Thailand, has adopted both empirical and mechanistic methods for flexible pavement analysis and design. Recently, the deflection-based design approach has been comprehensively reviewed by the DOH for the possible adoption of local design standards and practices. One of the key reasons was that Thailand's road authorities, i.e., the DOH and the Department of Rural Roads (DRR), have considered the FWD for the new construction and rehabilitation of road pavement. The FWD deflection measurement is a reliable method to assess the structural integrity and bearing capacity of pavement systems. In addition, the FWD has been widely accepted as a non-destructive test for deflection measurement and structural evaluation for long-term pavement performance. Ultimately, the implication of FWD deflections for in-house pavement analysis and design shall be developed and proposed to Thailand's road authorities. The objective of this paper is therefore to propose a new flexible pavement design approach based on FWD deflection measurement for practical adoption at a local level by Thailand's road authorities (the DOH and the DRR).

## 2. Background

Mechanistic pavement design involves calculating the response of the pavement to traffic loads using a mathematical model. The fundamental principle is similar to buildings and other structural designs. Equations used to calculate the deflections and strains result from loads imposed on columns and beams, while those used in the pavement system are a little more complicated but similar concepts. A series of charts and tables for calculating stresses and strains in pavement systems were developed in the 1950s. In the 1960s, computer programs became commercially available, but it was not until the development of personal computers in the late 1980s that agencies began to implement mechanistic pavement design. The main benefit of mechanistic design is not that it results in significantly different pavement structures, but that it considers the impact of materials, traffic, and the environment on their performance and service life.

Layered elastic analysis (LEA) has been widely adopted in most mechanistic pavement designs. It is commonly used to examine the responses of a multi-layered pavement structure. This LEA generally assumes pavement layers to be homogenous, isotropic, and linear elastic and to have finite thicknesses with a modulus of elasticity and Poisson's ratio. The applied vertical load is assumed to be uniformly distributed over a circular area. The corresponding structural responses, e.g., stress, strain, deflection, etc., can be determined through LEA. Several pieces of computer software for LEA have been developed and utilized in pavement analysis and design practices due to their simplicity. Examples of commercial LEA software include ILLI-PAVE, MICH-PAVE, DAMA, KENLAYER, CHEVRON,

BISAR, ELSYM5, VESYS, WESLEA, EVERSTRESS, CIRCLY, etc. Sawangsuriya et al. [9] utilized three-dimensional (3-D) finite element analysis (FEA) and LEA to examine the structural responses of flexible pavement under different types of axle group loads, e.g., a single axle-dual wheel, a tandem axle-dual wheel, and a tridem axle-dual wheel, and compared them with the field measurement data. The results indicated that both FEA and LEA were in good agreement with the field measurement results with some exceptions for strains under the asphalt surface. The structural responses in terms of vertical stresses, vertical strains, and horizontal strains from the LEA were identical to the FEA results. Thus, both FEA and LEA approaches could be applied for the estimation of pavement structural responses.

Flexible pavement surface deflection measurement is one of the direct testing methods of evaluating pavement response. There are of course other measurements that indicate a pavement's structural condition, but a surface deflection measurement provides the simplest and most direct alternative because the magnitude and shape of pavement deflection are functions of load characteristics, the pavement structure, and environmental conditions. In back-calculation analyses, deflection measurements can be applied to determine the stiffness of the pavement layers and the natural subgrade. Therefore, several pavement characteristics can be examined by directly measuring surface deflection under a specified load application.

## 3. Methodology

### 3.1. Pavement Structure and Material Properties

A total combination of 625 conventional flexible pavement sections was investigated in this study. Typical pavement layers used in Thailand are asphalt surfaces, crushed rock bases, soil-aggregate subbases, and selected materials above the natural subgrade, as shown in Figure 1. The five asphalt surface thicknesses included 50, 100, 150, 200, and 250 mm. The five crushed rock base thicknesses included 100, 150, 200, 250, and 300 mm. The five soil-aggregate subbase thicknesses included 100, 200, 300, 400, and 500 mm. The five selected material thicknesses included 0, 100, 200, 300, and 400 mm. It is noteworthy that the minimum thickness of the asphalt surface and aggregate base was 50 mm and 100 mm, respectively, according to AI [10] and AASHTO [11].

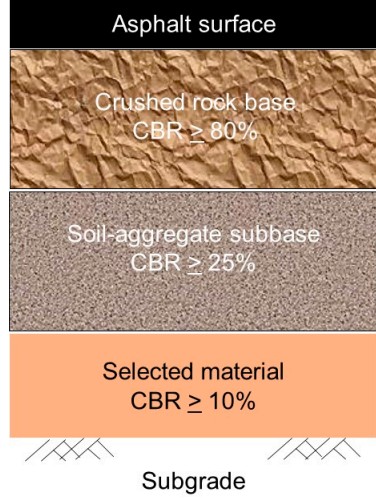

**Figure 1.** Typical pavement layers used in Thailand.

In this study, three methods of pavement structural thickness calculation, the AASHTO structural number (SN) [11], the Asphalt Institute (AI) full-depth asphalt thickness ($T_a$) [12], and the Odemark's method of equivalent thickness ($h_e$) [13] were utilized in order to classify these 625 sections of a flexible pavement structure. Table 1 summarizes four classes of these pavement sections based on AASHTO [11], AI [12], and Odemark's method [13].

Table 2 summarizes the elastic moduli and Poisson's ratios for the pavement layer materials according to the DOH's pavement design practice.

**Table 1.** Classification of the pavement structure.

| Class | SN | $T_a$ (cm) | $h_e$ (cm) |
|-------|------|--------|---------|
| 1 | 2–4 | 14–28 | 31–70 |
| 2 | 4–6 | 28–43 | 70–107 |
| 3 | 6–9 | 43–57 | 107–144 |
| 4 | 9–11 | 57–72 | 144–181 |

**Table 2.** Pavement layer material properties.

| Pavement Materials | Elastic Modulus (MPa) | Poisson's Ratio |
|--------------------|-----------------------|-----------------|
| Asphalt surface | 2500 | 0.35 |
| Crushed rock base | 350 | 0.35 |
| Soil-aggregate subbase | 150 | 0.35 |
| Selected material | 100 | 0.35 |
| Subgrade | 40 | 0.40 |

### 3.2. Load Applications

This study considered two types of load applications, e.g., an FWD and a standard Thai truck. According to the FWD test protocol adopted by the DOH's practice, a loading pressure ranging from 700 to 800 kPa is typically selected along with a plate radius of 0.15 m. A standard 10-wheel 25-ton truck is typically considered in Thailand for pavement design and analysis. This standard Thai truck with a tandem axle-dual wheel had a tire pressure of 690 kPa and a single axle load of 100 kN. A tire contact radius of 0.11 m was then determined by dividing a wheel load of 25 kN by a tire pressure of 690 kPa. It is noted that an FWD loading stress of 800 kPa was selected in this study because its corresponding deflection was closer to the 690 kPa standard Thai truck deflection when compared to the 700 kPa FWD deflection [14]. Table 3 summarizes the pressure and radius of contact for the FWD and the standard Thai truck.

**Table 3.** Pressure and radius of contact for the FWD and the standard Thai truck.

| Load Application | Pressure (kPa) | Radius of Contact (m) |
|------------------|----------------|------------------------|
| FWD | 800 | 0.15 |
| Truck | 690 | 0.11 |

### 3.3. Layered Elastic Analysis

A layered elastic analysis (LEA) was employed to determine the pavement responses under FWD and standard Thai truck load applications. The assumptions of Burmister's theory were considered as the following: (1) each pavement layer is homogeneous, isotropic, and linearly elastic, (2) the materials are weightless and infinite, (3) each layer has a uniform thickness and the subgrade layer is infinite, (4) uniform loading is applied on the surface over a circular area, (5) there are fully bonded interfaces between the layers, and (6) there are frictionless interfaces between the layers (e.g., no discontinuity of shear stress and radial displacement at each side of the interface). The essential input parameters were the material properties for each layer, i.e., elastic modulus and Poisson's ratio, layer thicknesses, load configurations, the number of load groups, and the x, y, and z coordinates for loads and responses.

Pavement responses under the FWD load application were determined right in the center of the loading plate, while the pavement responses under the standard Thai truck were determined at four positions, e.g., under the wheel load, between the dual-wheel load, between the tandem-axle load, and between the dual-wheel and tandem-axle load,

etc. Only the maximum surface deflections under these load applications ($d_o$) were considered and are reported herein. The standard Thai truck had a center-to-center spacing of 1300 mm between the axles and 330 mm between the wheels. Figure 2 illustrates four positions of pavement responses as well as the axle and wheel configurations of the standard Thai truck.

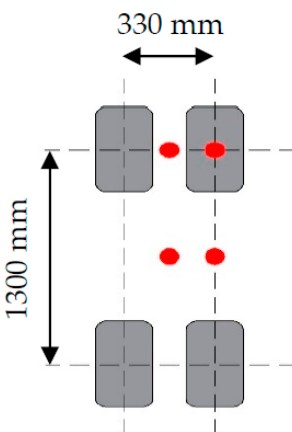

**Figure 2.** Four positions of pavement responses and the axle/wheel load configurations of the standard Thai truck.

*3.4. FWD Measurement*

The falling weight deflectometer (FWD) is the most common device for applying an impulse load to a pavement surface, and the corresponding deflection basin is measured by a series of sensors. The FWD in this study consisted of four main components: (1) an impulsive-force generator that enables the application of variable weights to the pavement, (2) a loading plate to transfer the impulsive force uniformly through the tested layer surface, (3) sensors for deflection basin determination, and (4) a data acquisition and processing system. Deflection data collected by a series of sensors were then used to calculate the pavement stiffness in terms of the layer modulus. During the testing operation, the vehicle was stopped, and the loading plate was placed directly over the test location. The sensor was lowered to the pavement surface, and the mass was subsequently dropped. The applied impulse load was varied depending on the height of the falling mass. A number of tests were conducted at the same location using different drop heights. The energy was transmitted through a circular plate to the pavement in a half-sine waveform of approximately 20 to 60 milliseconds. The deflection basin was detected and displayed on a computer screen inside the vehicle. Figure 3 illustrates a typical FWD deflection measurement.

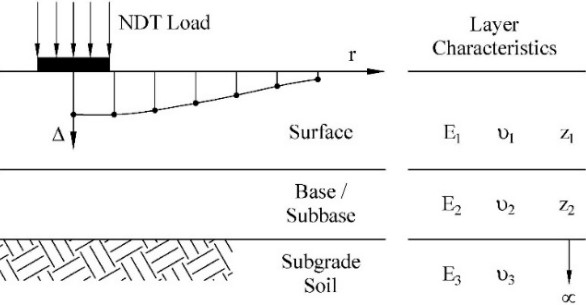

**Figure 3.** Typical FWD deflection measurement.

The FWD is commonly used in Thailand by the Department of Highways (DOH) and the Department of Rural Roads (DRR) for the structural condition evaluation of a pavement section, where the back-calculation of the subgrade and the pavement layer moduli is employed to characterize the structural condition. In particular, the Bureau of

Road Research and Development (BRRD), DOH, Thailand, has possessed the Dynatest Model 8000 FWD system from Denmark since 2000, as shown in Figure 4. The FWD was used to measure the surface deflections through nine surface sensors (geophones). A deflection bowl was generated by the impulse force, which was created by varying the drop height and weight. The sensors were located at 0, 200, 300, 450, 600, 900, 1200, 1500, and 1800 mm distances from the center of the loading plate. The load was transmitted to the road pavement through a 300 mm-diameter loading plate. The magnitude of the load was measured by a load cell. Note that the average pavement temperatures during the test ranged between 37 and 42 °C. A temperature correction of 40 °C was applied when evaluating the deflection. A number of research projects by the BRRD have involved the practical implications of FWD deflection measurements after the construction stage as well as during the in-service stage. Many of them are well-documented in final reports, refereed journal publications, and conference proceedings [15–19].

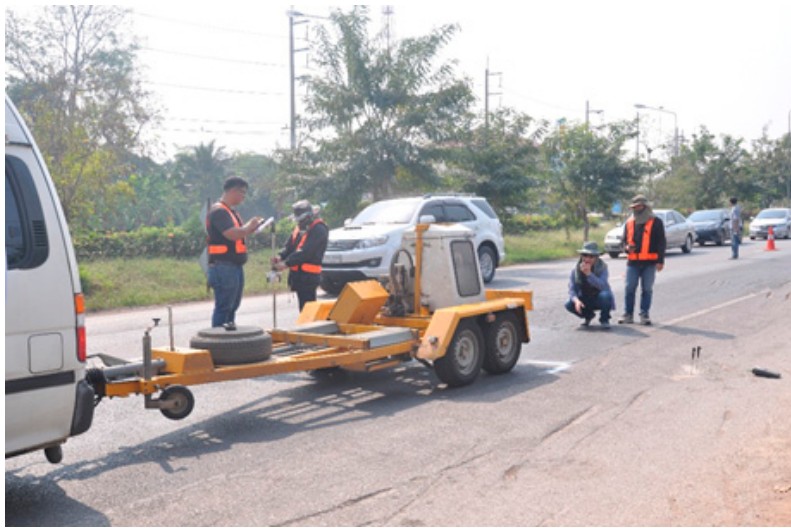

**Figure 4.** Dynatest Model 8000 falling weight deflectometer (FWD) from Denmark.

## 4. Results and Discussion

### 4.1. $d_o/d_{o,DOH}$–Pavement Structural Thicknesses Relationship

A layered elastic analysis (LEA) was used to determine the pavement responses in terms of surface deflection ($d_o$). The deflections from the FWD and standard Thai truck loads are presented herein. In this study, a normalized deflection ($d_o/d_{o,DOH}$) was introduced to eliminate the unit and to overcome the FWD measurement constraints, e.g., operation, configuration, model, etc. Such normalized deflection was defined as the maximum surface deflection ($d_o$) divided by the maximum surface deflection determined from the typical DOH pavement section ($d_{o,DOH}$). According to the Thai DOH's design practice, a typical pavement section consists of a 100 mm asphalt surface, a 200 mm crushed rock base, a 200 mm soil-aggregate subbase, and 200 mm of selected material. The relationships between normalized deflections and pavement structural thicknesses in terms of SN, $T_a$, and $h_e$ are illustrated in Figures 5–7. The results suggested that the relationships were divided into two sets of data: (1) a thin asphalt surface (AC) with a thickness of less than 100 mm and (2) a thick asphalt surface (AC) with a thickness of no less than 100 mm.

For the thin asphalt surface as shown in Figures 5a, 6a and 7a, the relationships between normalized deflections and pavement structural thicknesses in terms of SN, $T_a$, and $h_e$ were divided into two groups: group 1 with asphalt surface thicknesses of 50 mm and base thicknesses of 100 mm, and group 2 with asphalt surface thicknesses of 50 mm and base thicknesses between 150 and 300 mm. For the thick asphalt surfaces as shown in Figures 5b and 6b, the relationships between normalized deflections and pavement structural thicknesses in terms of SN, $T_a$, and $h_e$ were divided into five groups: group 1

with asphalt surface thicknesses of 100 mm and base thicknesses of 100 mm, group 2 with asphalt surface thicknesses of 100 mm and base thicknesses between 150 and 300 mm, group 3 with asphalt surface thicknesses of 150 mm and base thicknesses between 100 and 300 mm, group 4 with asphalt surface thicknesses of 200 mm and base thicknesses between 100 and 300 mm, and group 5 with asphalt surface thicknesses of 250 mm and base thicknesses between 100 and 300 mm. The parameter $d_o/d_{o,DOH}$ can be directly estimated from these graphs for a specified pavement structural thickness.

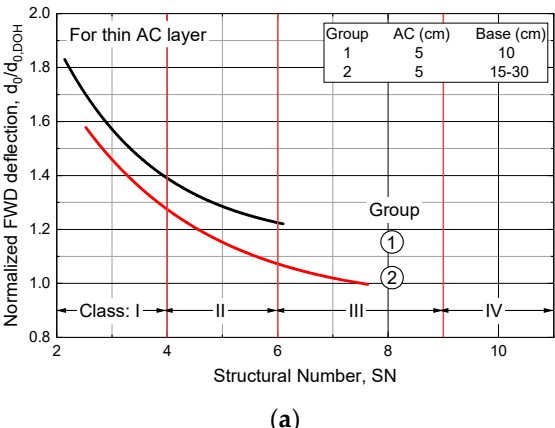

(**a**)

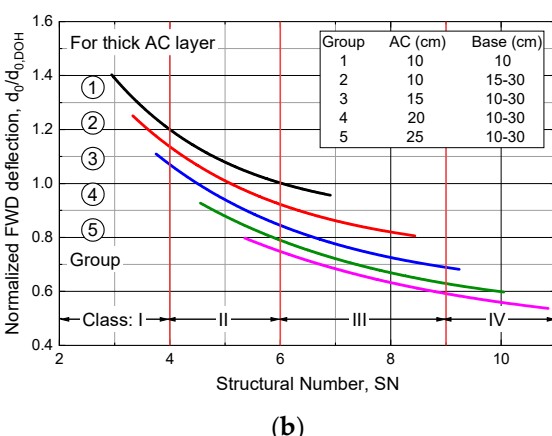

(**b**)

**Figure 5.** Normalized deflections vs. SN: (**a**) thin AC layer and (**b**) thick AC layer.

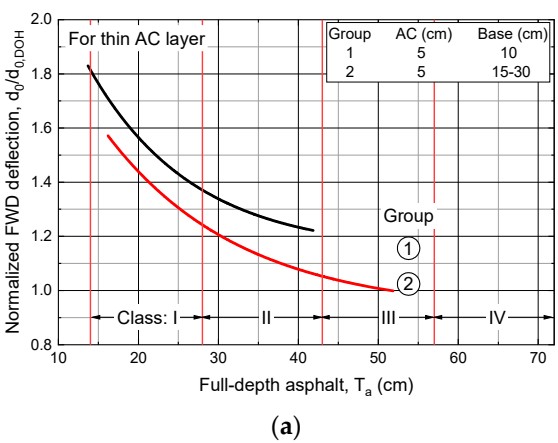

(**a**)

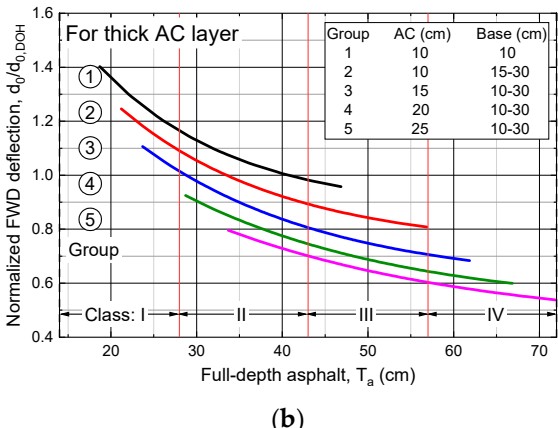

(**b**)

**Figure 6.** Normalized deflections vs. $T_a$: (**a**) thin AC layer and (**b**) thick AC layer.

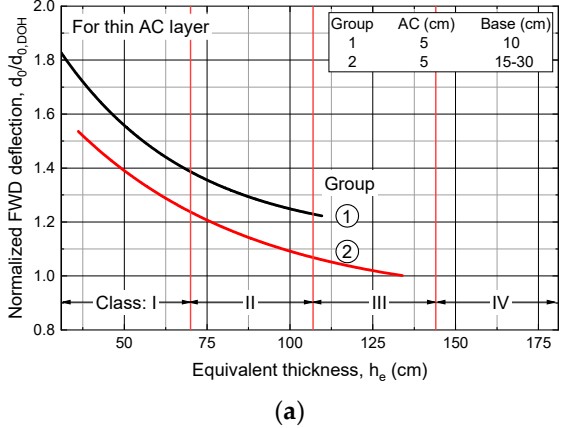

(**a**)

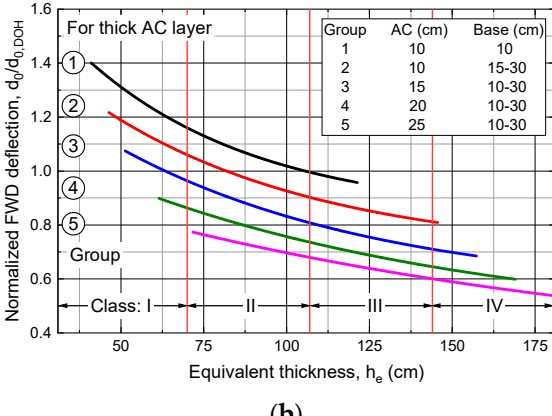

(**b**)

**Figure 7.** Normalized deflections vs. $h_e$: (**a**) thin AC layer and (**b**) thick AC layer.

### 4.2. Comparison between the $d_o/d_{o,DOH}$ from the LEA and the $d_o/d_{o,DOH}$ from the FWD

A comparison between the $d_o/d_{o,DOH}$ from the LEA and the $d_o/d_{o,DOH}$ from the FWD is carried out herein. In order to compare the $d_o/d_{o,DOH}$ from the LEA with the $d_o/d_{o,DOH}$ from the FWD, a total of eighteen road pavement sections as part of the national highway network were selected for the FWD measurements. Details of the FWD measurements on these road pavement sections as well as the layer thicknesses and their corresponding properties have been reported by the DOH [17]. Among these eighteen road pavement sections, there exist four pavement sections consisting of a 100 mm asphalt surface, a 200 mm crushed rock base, a 150 mm soil-aggregate subbase, and 150 mm of selected material, which were close to the typical DOH pavement section, i.e., a 100 mm asphalt surface, a 200 mm crushed rock base, a 200 mm soil-aggregate subbase, and 200 mm of selected material. The four road pavement sections therefore represent a typical DOH pavement section for this comparison, and the corresponding $d_{o,DOH}$ was then determined by averaging the $d_o$ of these four pavement sections.

Those parameters $d_o/d_{o,DOH}$ from the FWD measurements on the eighteen road pavement sections are summarized in Table 4. For a given road pavement section, the pavement structural thicknesses in terms of SN, $T_a$, and $h_e$ were calculated. It should be noted that since every section had a minimum asphalt surface thickness of 100 mm, only those relationships with thick asphalt surfaces were considered in this comparison study. Again, four pavement sections representing the typical DOH pavement section included highway No. 2 (ID 8), No. 331 (ID 11), No. 36 (ID 12), and No. 4 (ID 17). The parameters $d_o/d_{o,DOH}$ from the FWD were plotted on the $d_o/d_{o,DOH}$ from the LEA–pavement structural thickness relationship, as shown in Figure 8. Fourteen sections are presented in Figure 8 because four other pavement sections, e.g., highway No. 2 (ID 8), No. 331 (ID 11), No. 36 (ID 12), and No. 4 (ID 17), in Table 4 were considered as the typical DOH pavement section. Their deflection values were taken as the denominator for calculating a parameter $d_o/d_{o,DOH}$. It was observed that most parameters $d_o/d_{o,DOH}$ from the FWD were scattered along the curve-fitted relationship of $d_o/d_{o,DOH}$ from the LEA and pavement thickness, while some were close to the curve-fitted relationship. This suggested that although there existed some dispersions in the data, the implication of the $d_o/d_{o,DOH}$–pavement structural thickness relationship could be taken into account for the pavement design verification and performance evaluation.

**Table 4.** FWD measurements on the eighteen road pavement sections [17].

| ID | Highway No. | Section | SN | $T_a$ | $h_e$ | Average $d_o$ | $d_{o,DOH}$ | $d_o/d_{o,DOH}$ |
|----|-------------|---------|----|----|----|----|----|----|
| 1 | 32 | 119 + 800–120 + 050 | 5.9 | 37.4 | 82.1 | 444.6 | 228.3 | 2.02 |
| 2 | 35 | 54 + 000–54 + 250 | 6.4 | 38.9 | 83.6 | 212.0 | 228.3 | 0.96 |
| 3 | 1 | 440 + 000–440 + 250 | 6.4 | 42.2 | 108.8 | 56.6 | 228.3 | 0.26 |
| 4 | 1 | 440 + 750–441 + 000 | 6.4 | 42.2 | 108.8 | 49.5 | 228.3 | 0.22 |
| 5 | 117 | 24 + 750–25 + 000 | 7.0 | 44.4 | 110.5 | 141.9 | 228.3 | 0.64 |
| 6 | 1 | 308 + 300–308 + 550 | 5.9 | 37.4 | 82.1 | 458.2 | 228.3 | 2.08 |
| 7 | 1 | 379 + 950–380 + 200 | 6.4 | 42.2 | 108.8 | 213.2 | 228.3 | 0.97 |
| 8 | 2 | 268 + 100–268 + 350 | 4.6 | 30.6 | 76.4 | 185.6 | 228.3 | 0.84 |
| 9 | 24 | 98 + 750–99 + 000 | 6.2 | 42.6 | 110.6 | 152.3 | 228.3 | 0.69 |
| 10 | 3 | 54 + 250–54 + 500 | 4.3 | 27.4 | 61.7 | 301.9 | 228.3 | 1.37 |
| 11 | 331 | 47 + 060–47 + 300 | 4.6 | 30.6 | 76.4 | 172.2 | 228.3 | 0.78 |
| 12 | 36 | 28 + 575–28 + 825 | 4.6 | 30.6 | 76.4 | 303.0 | 228.3 | 1.38 |
| 13 | 323 | 4 + 900–5 + 150 | 6.9 | 46.1 | 117.3 | 431.5 | 228.3 | 1.96 |
| 14 | 4 | 473 + 500–473 + 750 | 5.5 | 37.2 | 96.4 | 83.7 | 228.3 | 0.38 |
| 15 | 4 (LW) | 88 + 570–88 + 770 | 6.9 | 45.7 | 114.4 | 311.4 | 228.3 | 1.41 |
| 16 | 4 (RW) | 88 + 570–88 + 770 | 6.9 | 45.7 | 114.4 | 284.5 | 228.3 | 1.29 |
| 17 | 4 | 304 + 750–305 + 000 | 4.6 | 30.6 | 76.4 | 252.6 | 228.3 | 1.15 |
| 18 | 4 | 170 + 750–171 + 000 | 5.6 | 37.6 | 95.6 | 296.8 | 228.3 | 1.30 |

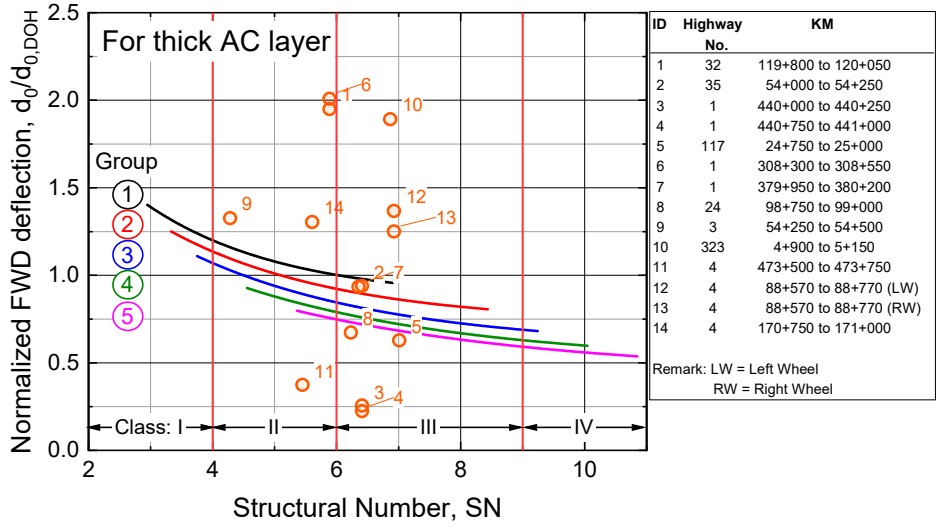

(**a**) $d_o/d_{o,DOH}$ vs. SN.

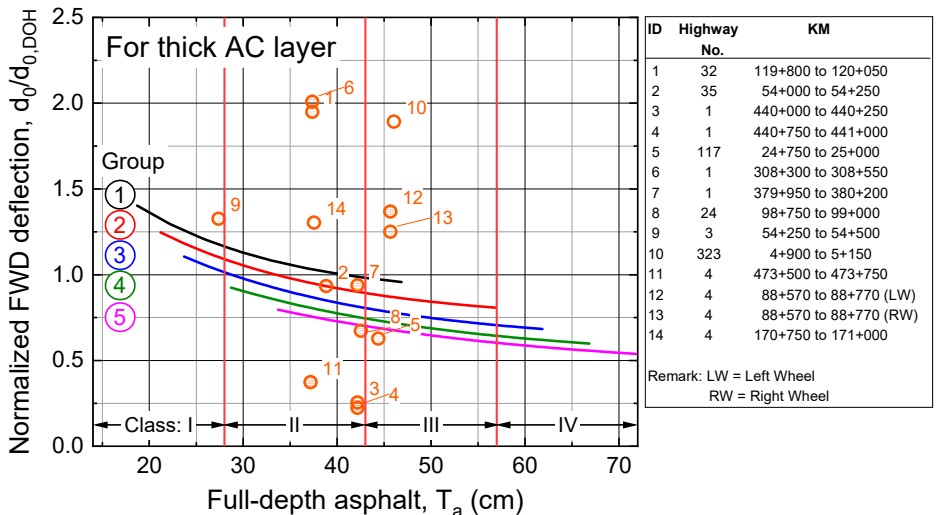

(**b**) $d_o/d_{o,DOH}$ vs. $T_a$.

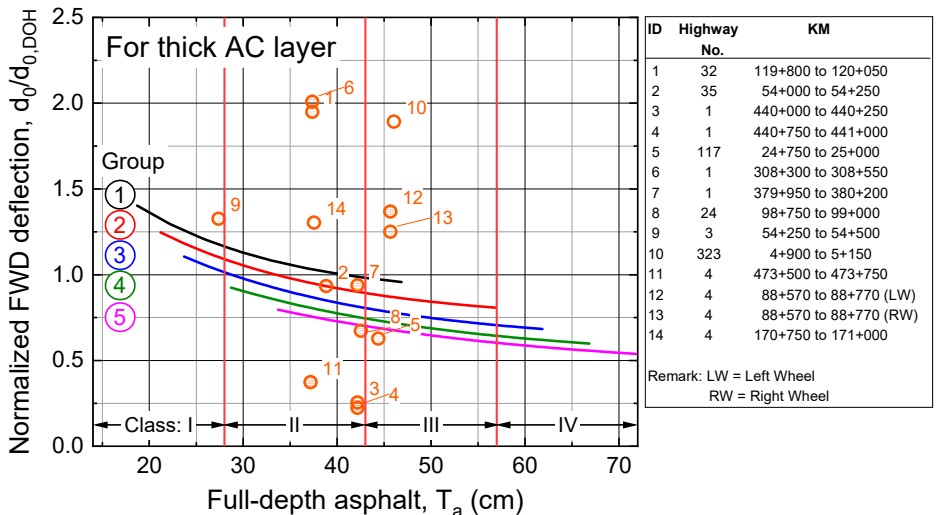

(**c**) $d_o/d_{o,DOH}$ vs. $h_e$

**Figure 8.** Comparison between the $d_o/d_{o,DOH}$ from the LEA and the $d_o/d_{o,DOH}$ from the FWD.

## 5. Conclusions

Thailand's DOH has adopted both empirical and mechanistic approaches for flexible pavement analysis and design. LEA has been widely adopted in most mechanistic designs. It assumes pavement layers to be homogenous, isotropic, and linear elastic and to have a finite thickness with a modulus of elasticity and Poisson's ratio. The deflection-based design approach has recently been reviewed by the DOH for the possible adoption of local design standards and practices. Since the FWD has been widely accepted as the non-destructive test for deflection measurement and structural integrity as well as bearing capacity evaluation for the long-term performance of pavement systems, Thailand's road authorities have considered the FWD for the new construction and rehabilitation of road pavements.

This study compared the calculated $d_o$ from the LEA and the measured $d_o$ from the FWD. A normalized deflection (e.g., the ratio of $d_o$ to $d_o$ from a typical DOH pavement section, $d_o/d_{o,DOH}$) was introduced and was plotted against the pavement structural thickness in terms of SN, $T_a$, and $h_e$. The $d_o/d_{o,DOH}$ from the LEA decreased as the pavement structural thickness in terms of SN, $T_a$, and $h_e$ increased, with two isolated trends: one for thin (<100 mm thick) and another for thick (>100 mm thick) asphalt surfaces (AC layers). A total of eighteen road pavement sections as part of the national highway network were selected for the actual FWD measurements. The comparison results suggested that most parameters $d_o/d_{o,DOH}$ from the FWD measurements were scattered along the curve-fitted relationship of $d_o/d_{o,DOH}$ from the LEA and pavement thickness, while some were close to the curve-fitted relationship. This suggested that although there existed some dispersions in the data, the implication of the $d_o/d_{o,DOH}$–pavement structural thickness relationship could be taken into account for pavement design verification and performance evaluation. The purpose of this implication is to check if the actual pavement deflections after the project completion are consistent with the theoretical design values. Based on this proposed conceptual framework, the long-term pavement deflections can also be evaluated by comparing them with their designed deflection benchmark. This proposed method, however, requires additional field measurement verification prior to practical implementation.

**Author Contributions:** Author Contributions: writing—original draft preparation, A.S.; writing—review and editing, T.S. and P.J. All authors have read and agreed to the published version of the manuscript.

**Funding:** This research was funded by Thailand Science Research and Innovation (TSRI).

**Institutional Review Board Statement:** Not applicable.

**Informed Consent Statement:** Not applicable.

**Data Availability Statement:** Not applicable.

**Acknowledgments:** This research project was funded by Thailand Science Research and Innovation, Ministry of Higher Education, Science, Research and Innovation. The technical support from the Department of Highways, Ministry of Transport, is greatly appreciated.

**Conflicts of Interest:** The authors declare no conflict of interest.

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
