# Peer review of "Deflection-Based Approach for Flexible Pavement Design in Thailandâ€"

_infrastructures, doi:10.3390/infrastructures8070116_

Round 1
Reviewer 1 Report
The introduction is incomplete and it would be worthwhile to study the international literature in more detail.
The description of the method used is not detailed enough, e.g. what temperature correction was applied when evaluating the deflection?
There are 18 road sections in the text, 17 in Table 4 and 14 in Figure 6, why?
The support for the conclusion is incomplete, the dispersion of the data is quite large and does not support the authors' optimistic conclusions.
Author Response
The introduction is incomplete and it would be worthwhile to study the international literature in more detail.
The new statements with citations and references have been added as follows:
The FEA has been widely adopted in the recently developed mechanistic design and performance analysis of the pavement system because of its versatile implication of mechanical characterization [1] – [5]. The LEA is however practically adopted because of its simplicity and cost of analysis.
- Cho, Y., Mccullough, B.F. and Weissmann, J. Consideration on finite element method application in pavement structural analysis. Transportation Research Record, 1996, No. 1539, pp. 96-101.
- Kim, M. Three-dimensional finite element analysis of flexible pavement considering nonlinear pavement foundation be-havior.” Ph.D. Dissertation, University of Illinois at Urbana-Champaign, IL, USA, 2007.
- Ban, H., Im, S. and Kim, Y. Truck loading on design and analysis of asphaltic pavement structures. Mid-America Trans-portation Center, 2010.
- Haselbach, L., William, F. and Alam, C.A. Three-dimensional finite element modelling of pervious concrete pavement: ver-tical porosity destribution approach. International Journal of Research in Engineering and Technology, 2013; Vol. 12, No. 2, pp. 767-777.
- Zhu, C., Li, X., Pei, J., Chen, J. and Zhang, J. Characterizing the three-stage rutting behavior of asphalt pavement with semi-rigid base by using UMAT in ABAQUS. Construction and Building Materials, 2017, Vol. 140, pp. 496-507.
The description of the method used is not detailed enough, e.g. what temperature correction was applied when evaluating the deflection?
The following statement has been added. The average pavement temperatures during the test ranged between 37 and 42 oC. A temperature correction of 40 oC was applied when evaluating the deflection.
There are 18 road sections in the text, 17 in Table 4 and 14 in Figure 6, why?
Table 4 has been corrected to 18 sections. Thank you. The following statements have been added.
Fourteen sections were presented in Figure 7 because four other pavement sections e.g., highway No. 2 (ID 8), No. 331 (ID 11), No. 36 (ID 12), and No. 4 (ID 17) in Table 4 were considered as the typical DOH pavement section. Their deflection values were taken as the denominator for calculating a parameter do/do, DOH.
The support for the conclusion is incomplete, the dispersion of the data is quite large and does not support the authors' optimistic conclusions.
The revised statement has been incorporated in the conclusion.
This suggested that although there existed some dispersions in the data, the implication of do/do, DOH - pavement structural thickness relationship could be taken into account for the pavement design verification and performance evaluation. The purpose of this implication is to check if the actual pavement deflections after the project completion are in consistent with the theoretical design values. Based on this proposed conceptual framework, the long-term pavement deflections can be also evaluated by comparing with their designed deflection benchmark. This proposed method however requires additional field measurement verification prior to the practical implication.

Reviewer 2 Report
The paper is interesting with potentially local implications, but in order to be considered for publication in a journal it must be extended in length and scientific impact factor. Please find below my comments.
1_ Line 112-113: Which layer is used? Subgrade? Please reconsider.
2_ Please provide a typical pavement cross-section. The paper needs to become more illustrative.
3_ Please justify the selected moduli in table 2. Use references from other studies to strengthen your views.
4_ Lines 139-148 present no new knowledge. You can simply state that the assumptions of Burmister’s theory were considered.
5_ Please explain the practical significance of a sensitivity analysis for pavement design practices. Any recommendations for road types, etc.
6_ My main concern about this paper is the need to drastically improve the scientific content of the paper with critical discussion points, recent references and the authors’ perspectives. References are limited (especially for a journal paper). For example, the elastic theory exhibits some limitations especially for AC materials because of the AC viscoelastic nature. There have been attempts to consider this aspect in pavement design and maintenance. The FWD is indeed a tool that is used for many pavement applications especially in terms of monitoring aspects. There are also ways to combine the FWD testing with viscoelastic analysis according to the MEPDG. Such aspects, although not being the main aim of the study, should be at least discussed in the introduction. Please consult other studies as well, e.g., you may see and cite https://doi.org/10.3390/infrastructures7050061 , https://doi.org/10.1080/10298436.2021.1995733 , or other similar ones. Please reconsider.
Overall, pavement design is an important issue and ongoing improvements are discussed especially because of new loading patterns of moving vehicles, new distress types, etc. Therefore, the paper needs to be reconsidered once it is drastically revised by the authors.
Meaningful, but a moderate editing is required.
Author Response
The paper is interesting with potentially local implications, but in order to be considered for publication in a journal it must be extended in length and scientific impact factor. Please find below my comments.
1_ Line 112-113: Which layer is used? Subgrade? Please reconsider.
The authors are not sure if they understand the question correctly. Typical pavement system in Thailand consists of asphalt surface, crushed rock base, soil-aggregate subbase, and selected material above natural subgrade. In this study, the pavement system was varied to achieve 625 combinations. Five asphalt surface thicknesses included 50, 100, 150, 200, and 250 mm. Five crushed rock base thicknesses included 100, 150, 200, 250, and 300 mm. Five soil-aggregate subbase thicknesses included 100, 200, 300, 400, and 500 mm. Five selected material thick-nesses included 0, 100, 200, 300, and 400 mm.
2_ Please provide a typical pavement cross-section. The paper needs to become more illustrative.
A typical pavement layers used in Thailand has been added in Figure 1.
3_ Please justify the selected moduli in table 2. Use references from other studies to strengthen your views.
These pavement layer moduli were specified in accordance with the Department of Highways’s pavement design practice.
4_ Lines 139-148 present no new knowledge. You can simply state that the assumptions of Burmister’s theory were considered.
The new statement has been added as per recommendation.
5_ Please explain the practical significance of a sensitivity analysis for pavement design practices. Any recommendations for road types, etc.
6_ My main concern about this paper is the need to drastically improve the scientific content of the paper with critical discussion points, recent references and the authors’ perspectives. References are limited (especially for a journal paper). For example, the elastic theory exhibits some limitations especially for AC materials because of the AC viscoelastic nature. There have been attempts to consider this aspect in pavement design and maintenance. The FWD is indeed a tool that is used for many pavement applications especially in terms of monitoring aspects. There are also ways to combine the FWD testing with viscoelastic analysis according to the MEPDG. Such aspects, although not being the main aim of the study, should be at least discussed in the introduction. Please consult other studies as well, e.g., you may see and cite https://doi.org/10.3390/infrastructures7050061 , https://doi.org/10.1080/10298436.2021.1995733 , or other similar ones. Please reconsider.
The new statements with citations and references have been added as follows:
It is noteworthy that the elastic theory exhibits some limitations especially for asphalt materials because of their viscoelastic nature. There have been attempts to consider this aspect in pavement design and maintenance. The FWD is indeed a tool that is used for many pavement applications especially in terms of monitoring aspects. Although, there are also approaches to combine the FWD testing with viscoelastic analysis [6] – [8], such aspects were not the main aim of the study.
- Al-Khateeb, L.A., Saoud, A. and Al-Msouti, M.F. Rutting prediction of flexible pavements using finite element modeling. Jordan Journal of Civil Engineering, 2011, Vol. 5, No. 2, pp. 173-190.
- Gkyrtis, K., Plati, C. and Loizos, A. Mechanistic analysis of asphalt pavements in support of pavement preservation deci-sion-making. Infrastructures, 2022, Vol. 7, No. 5, pp. 61.
- Gkyrtis, K., Armeni, A. and Loizos, A. A mechanistic perspective for airfield pavements evaluation focusing on the asphalt layers’ behaviour. International Journal of Pavement Engineering, 2022, Vol. 23, No. 14, pp. 5087-5100.
Overall, pavement design is an important issue and ongoing improvements are discussed especially because of new loading patterns of moving vehicles, new distress types, etc. Therefore, the paper needs to be reconsidered once it is drastically revised by the authors.

Round 2
Reviewer 1 Report
- with corrections the article is acceptable
Author Response
The authors wish to thank Reviewer 1 for every comment and suggestion.

Reviewer 2 Report
The paper was improved and my comments were replied. However, there is still margin for improvements with the most important one being the objective statement of this paper. In other words, what are the study's innovation, either in a local level or in an international level. Please elaborate further at the end of the introduction.
Moderate editing is needed.
Author Response
The paper was improved and my comments were replied. However, there is still margin for improvements with the most important one being the objective statement of this paper. In other words, what are the study's innovation, either in a local level or in an international level. Please elaborate further at the end of the introduction.
The authors wish to thank every instructive comment. Below is the response to Reviewer 2's comment.
The end of the introduction was revised as follows: "The objective of this paper is therefore to propose a new flexible pavement design approach based on the FWD deflection measurement for practical adoption in a local level by Thailand road authorities (DOH and DRR)."
